# Five-year fertilization alters soil microbial composition and functionality in sandy grassland

Rui Zhang,[1,2,3,4] Yulin Li,[1,3] Xueyong Zhao,[1,2] A. Allan Degen,[5] Xinping Liu,[1,3] Jie Lian,[1,3] Yuqiang Li,[1,3] Yalin Wu,[1,2,3] Zhanhuan Shang[6]

**ABSTRACT** The impacts of reclamation and fertilization of sandy grassland on soil microbial communities and functional groups related to carbon (C) and nitrogen (N) cycling are not well understood. To fill this gap, three types of fertilizers, namely, chemical fertilizer (CF), manure (M), and chemical fertilizer plus manure (CF_M), were applied annually for five years to reclaimed sandy cropland planted to maize. Nearby sandy grassland without fertilizer and maize was included as a control. Soil microbial communities and processes, soil properties, and aboveground biomass (AGB) were determined. Soil microbial Chao richness was lowest in soil without fertilizer and maize. Fungal Shannon diversity was lowest with chemical fertilizer plus manure, while soil microbial Chao richness and bacterial Shannon diversity were not influenced by fertilization. Reclamation and fertilization increased AGB, which was greatest with chemical fertilizer plus manure and was more than seven times greater than that of sandy grassland. Soil extracellular enzyme activities increased with chemical fertilizer plus manure. Fertilization enhanced C cycle functional groups by decreasing soil bulk density and elevating soil total N, total carbon, Firmicutes abundance, and bacterial Chao richness, but lessened N cycle functional groups by decreasing Nitrospirota abundance. Microbial functional category groups associated with C and N cycles responded differently to reclamation and fertilization of sandy soil, which, in turn, affected soil carbon sequestration and nutrient availability.

**IMPORTANCE** Reclamation and fertilization of sandy grassland altered biogeochemical functions by influencing microbial communities and functional category groups related to carbon (C) and nitrogen (N) cycling. Reclamation and fertilization could lead to the reduction of soil C content and insufficient soil N by altering functional category groups, which would be a potential risk leading to sandy grassland degradation. These findings not only improve our understanding of the consequences of sandy grassland reclamation and fertilization on ecosystem processes, but are also important for predicting soil C sequestration and nutrient cycling and for developing strategies to prevent degradation of sandy grassland.

**KEYWORDS** soil microbiome, community composition, C cycling, N cycling, environmental factors

Soil microorganisms play a pivotal role in organic matter formation and decomposition, nutrient cycling and supply, and plant productivity (1–3), and are indicators of soil health status (4). Bacterial species mediate the cycling of nutrient elements and, thus, influence soil quality (5), while fungal species are vital in symbiosis, decomposition, and pathogenesis (6). Consequently, soil microbial communities are essential for biogeochemical cycles and for ecosystem structure and functions (7–9).

Crop production generally depends on fertilizers to increase the yield. However, excessive application of chemical fertilizers (CFs) causes ammonia volatilization,

**Peer Reviewer** Saidu Abdullahi, Universiti Sains Malaysia, Pinang, Malaysia

Address correspondence to Yuqiang Li, liyq@lzb.ac.cn.

The authors declare no conflict of interest.

10.1128/spectrum.02963-25 **1**

denitrification, air pollution, and soil acidification and reduces the quality of agricultural products (10). Organic fertilizers have longer-lasting effects on fertility and contain more comprehensive nutrients than chemical fertilizers, which improve soil microbial community composition, fertility, and the quality of agricultural products (11). The application of organic and chemical fertilizers enhanced soil oxygen and nutrient availabilities largely due to the nitrification of microorganisms (12, 13). Consequently, fertilization can increase plant productivity and affect the soil microbial composition and function and, ultimately, influence soil organic carbon (C) dynamics and nutrient cycling (14, 15), especially nitrogen (N) cycling. For example, long-term chemical fertilizers reduced the turnover of microbial communities involved in soil nitrogen fixation and denitrification (16), whereas organic fertilizers increased the soil bacterial diversity and altered the community structure. The relative abundances of Firmicutes, Proteobacteria, and Zygomycota, and the N cycle-related soil microorganisms (diazotrophs, ammonia-oxidizers, and denitrifiers) increased with organic fertilizers (17, 18). Furthermore, soil fungi were reported to be important contributors to carbon and nitrogen turnover following the addition of organic fertilizers (19).

Sandy grassland crop cultivation is important globally for the livelihood of residents in arid and desert areas. However, sandy soils are fragile and are characterized by poor nutrients and low organic matter content (20). To improve the productivity of sandy grassland, numerous management strategies have been practiced. The reclamation of sandy grassland and the application of fertilizer to achieve greater crop yields have been gaining popularity.

Fertilization affects soil organic carbon and nutrient turnovers by altering soil microbial growth and influencing extracellular enzyme activity (21, 22). Soil bacteria were altered by edaphic properties changes that were induced by either fertilization (12) and/or land-use change (23). The development of agriculture in sandy grassland not only produces agricultural products, but also increases vegetation coverage, reduces sandstorms, improves air quality and resource utilization, protects the ecological environment, and, ultimately, promotes the economic status and stability of the region. Therefore, understanding the responses of soil structure, microorganisms, and cycling of nutrient elements to fertilization is important for maximum crop production in sandy grassland. However, the impact of fertilization on the composition, functionality, and biodiversity of soil microbial communities following the conversion of sandy grassland to maize cultivation remains poorly understood. The aim of this study was to fill this important gap. We hypothesized that land reclamation and fertilizers increase C cycle and decrease N cycle functional groups by altering soil properties and microbial community composition. To test this hypothesis, we determined soil microbial diversities and community compositions, soil extracellular enzyme activities, and potential functional category groups related to C and N cycles in a five-year fertilization study in a cropland growing maize, which was converted from a sandy grassland. The objectives were to examine: (i) the impacts of reclamation and fertilization on soil physico-chemical and biological properties, aboveground biomass (AGB), and soil microbial diversities and community structures; and (ii) the drivers of potential functional category groups related to C and N cycles. Results should be beneficial for the development of management strategies for sustainable crop production in sandy grassland.

## MATERIALS AND METHODS

### Study site

The study site, at Naiman Banner (42°14′–43°32′N, 120°19′–121°35′E) in the Horqin Sandy Land, is situated in the agro-pastoral ecotone of northern China. Multiple ecosystems co-exist in the area, including mainly sandy grassland and cropland. Nearly 80% of the sandy soil has undergone severe desertification due to wind erosion (24). Maize (*Zea mays* L.) production in irrigated cropland converted from grassland is the main agricultural activity in this region (25) and is considered to be an effective way to

control desertification. In recent decades, the converted cropland has expanded rapidly due to the growth in population (26). The area has a continental semi-arid monsoon temperate climate, with an average annual temperature of 5.8–6.4°C and a mean annual precipitation of 360 mm, which is concentrated from May to September. The soil is dominated by chestnut soil (27).

## Fertilization treatments after reclamation of natural sandy grassland for maize production

A randomized complete block design consisting of five blocks, each containing four experimental plots (11 m × 14 m) representing different treatments, was established in March 2018. The treatments were: no fertilizer (CK), chemical fertilizer (CF), manure (M), and chemical fertilizer plus manure (CF_M) (Table S1). Five sandy grassland plots (1 m × 1 m) near the maize fields, with no fertilizer, were selected as non-treated controls (SG). Fertilization was initiated in 2018, and application rates of CF and M were based on prevailing rates, which met the maize growth nutrient requirements in sandy cropland. Fertilization treatments are presented in Table S1.

## Sampling and soil parameters

Soil and plant samples were collected in the middle of August 2022, in the maize filling stage. A soil core at a depth of 0–20 cm was collected from each plot using a soil auger of 3 cm in diameter. To prevent cross-contamination between plots, the auger was sterilized with 75% ethanol solution between samples. The soil cores were homogenized and sieved through a 2 mm screen to remove debris. A 10 g subsample of each plot was stored at −80°C for DNA extraction, a 5 g subsample was stored at 4°C to measure extracellular enzyme activities, a 25 g subsample was used to determine water content, and the remaining soil was dried at room temperature to measure soil physicochemical characteristics. Aboveground biomass was calculated as the product of the dry mass per maize plant and planting density. Soil water content (SWC) was determined by oven-drying at 105°C for 48 h. Total carbon and nitrogen contents were measured using an elemental analyzer (Costech ECS4010, Milan, Italy). Soil pH and electrical conductivity (EC) were determined using a digital multiparameter instrument (PHS-3C, Puchun, Shanghai, China) (28). Extracellular enzyme activities associated with nutrient cycling were quantified by specifically targeting β-1,4-glucosidase (BG) for carbon acquisition, β-1,4-N-acetyl-glucosaminidase (NAG) and leucine aminopeptidase (LAP) for nitrogen metabolism, and alkaline phosphatase (AP) for phosphorus mobilization (29, 30). The activities of the extracellular enzymes were examined by standard fluorometric techniques (31, 32).

## Sequencing and bioinformatics analysis

The Mag-Bind Soil DNA Kit (Omega Bio-tek, Norcross, GA, USA) was used to extract soil genomic DNA. The concentration and the purity of the extracted DNA were determined by the 260/280 nm ratio (1.8–2.2) using a NanoDrop 2000 UV-vis Spectrophotometer (Thermo Scientific, Wilmington, DE, USA). Absolute abundances (the gene copy numbers) of the bacterial and fungal communities were estimated using quantitative real-time polymerase chain reaction (qPCR) amplification techniques with a Line-Gene 9600 Plus Cycler (Thermo Fisher Scientific Inc., Waltham, MA, USA). PCR amplification used the universal primer 338F-806R for bacteria (33) and SSU0817F-1196R for fungi (34) with a GeneAmp 9700 PCR system (Applied Biosystems, Foster City, CA, USA). PCR products were quantified using the QuantiFluorTM-ST Fluorometer (Promega Biotech, Beijing, China), and sequencing (by Majorbio Company, Shanghai, China) used an Illumina MiSeq platform (San Diego, USA). The raw reads of sequencing were quality-filtered by Fastp, merged by FLASH, and chimeric sequences were removed (35). The number of reads for all bacteria samples ranged between 34,356 and 45,469 and for all fungi samples, ranged between 38,549 and 49,231 (Fig. S1). Sequences

were grouped into the same operational taxonomic units (OTUs) with high similarity (≥97%) using UPARSE version 11 (36). The representative sequences for each OTU were annotated and classified based on the SILVA database (http://www.arb-silva.de) using the RDP (Ribosomal Database Project) classifier (https://sourceforge.net/projects/rdp-classifier/). The functional genes were predicted by FAPROTAX for bacteria and FUNGuild for fungi using the Kyoto Encyclopedia of Genes and Genomes (KEGG) (https://www.kegg.jp/kegg/) and National Center for Biotechnology Information Non-redundant (NCBI NR) (https://www.ncbi.nlm.nih.gov/) databases (37, 38) based on OTU representative sequences.

## Statistical analyses

Soil microbial diversity indices were calculated by Mothur software (version v.1.30.2). A one-way ANOVA, followed by Duncan's *post hoc* tests (SPSS Inc., Chicago, IL, USA) compared soil physico-chemical and aboveground biomass, soil microbial diversities and compositions, and microbial processes (soil extracellular enzyme activities and predicted functional category groups) among treatments. A level of ($P < 0.05$) was accepted as significant. Specific biomarkers of the treatments were classified by linear discriminant analysis (LDA) with LEfSe (39). The relationships between soil microbial processes and soil properties, aboveground biomass, soil microbial absolute abundances, and microbial richness were tested by Pearson's correlation coefficients. The principal coordinate analysis (PCoA) of the bacterial and fungal communities, based on the Bray-Curtis distance matrix, used the "vegan" package in R. Structural equation models (SEMs) determined the direct and indirect impacts of fertilizers on microbial C and N processes using the function of psem () in the piecewise SEM package in R (V.4.3.2).

## RESULTS

### Soil properties and aboveground biomass

Soil water content (SWC) was higher ($P < 0.05$) in M and CF_M than in SG; soil pHs of SG and CF were lower ($P < 0.05$) than the other treatments; while soil electric conductivity (EC) was lower ($P < 0.05$) in CK and CF than in SG and CF_M. Soil bulk density (SBD) of CF_M was lower ($P < 0.05$) than in the other treatments, except for M. Total soil carbon (SC), soil nitrogen (TN), and soil available nitrogen (AVN) were higher ($P < 0.05$) in M and CF_M than in the other treatments; and soil available phosphorus (AVP) was lower ($P < 0.05$) in CK than in M and CF_M. AGB in CK, CF, M, and CF_M was 7, 13, 12, and 17 times greater than in SG (Table S2).

The 16S rRNA gene copies (16S quantity) in CF_M were greater ($P < 0.05$) than in SG and CK, whereas the 18S rRNA gene copies (18S quantity) in CF_M were greater ($P < 0.05$) than in the other treatments, except for M (Fig. S2).

### Soil microbiome composition

Soil microbial Chao richness in SG was lower ($P < 0.05$) than in the other treatments, whereas the fungi Shannon diversity index in CF_M was lower ($P < 0.05$) than in the other treatments, except for CF (Fig. S3).

The dominant phyla for soil bacteria were Proteobacteria, Actinobacteriota, Acidobacteriota, and Firmicutes and for fungi were Ascomycota, Mucoromycota, and Basidiomycota (Fig. S4). The bacterial phyla Actinobacteriota, Firmicutes, Bacteroidetes, Gemmatimonadetes, Nitrospirota, Cyanobacteria, unclassified_k_noank_d_Bacteria, Patescibacteria, Bdellovibrionota, Methylomirabilota, and the fungal phyla SAR_k_norank, Chytridiomycota, Schizoplasmodiida, unclassified_d_Eukaryota, and Aphelidea differed significantly among treatments (Fig. 1). The relative abundances of Gemmatimonadaceae and Rubrobacter increased ($P < 0.001$) with reclamation, but those of Knufia and Verticillium decreased ($P < 0.05$) with reclamation and fertilizer. The relative abundances of Sporosarcina, Paenibacillus, Steroidobacter, and Turicibacter increased ($P < 0.05$), but those of Geodermatophilaceae, Rubrobacter, and Nitrospira

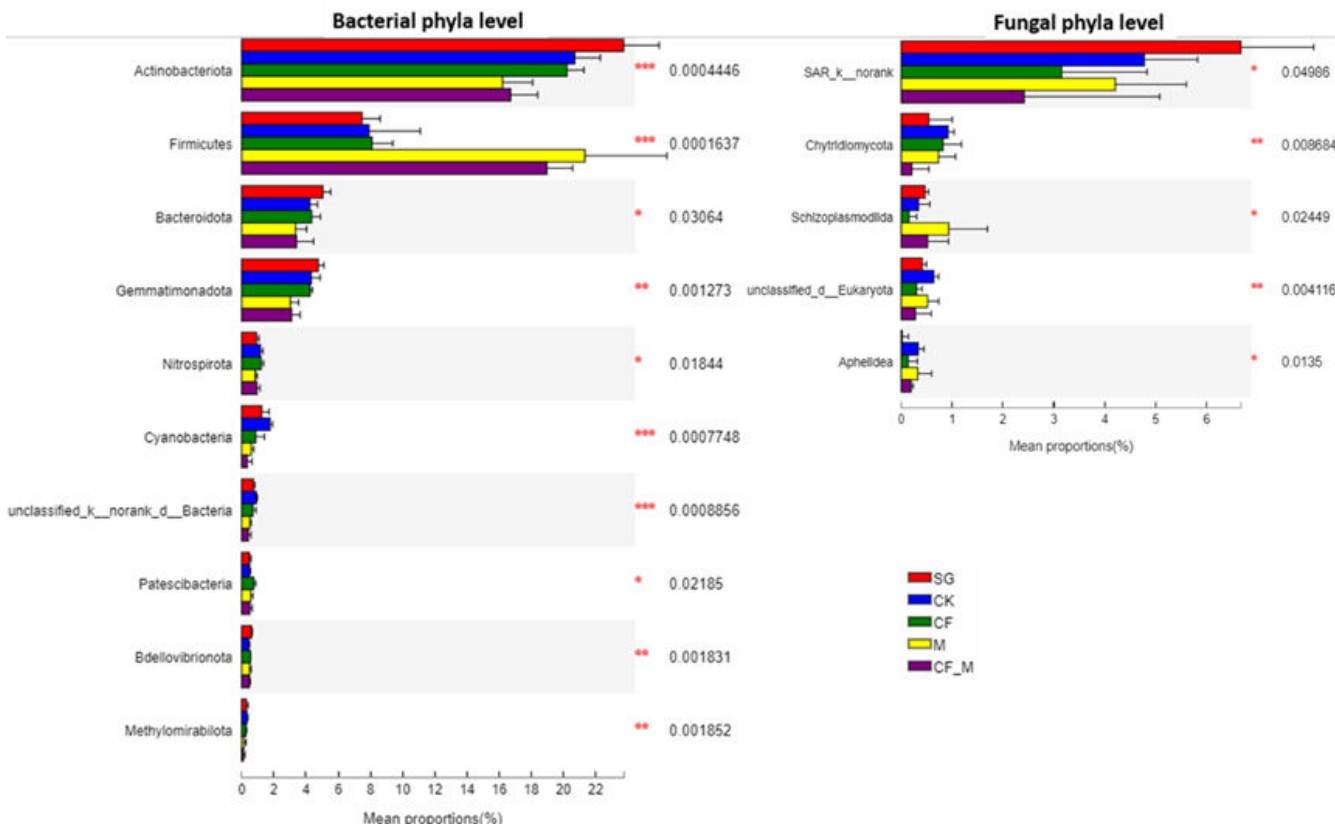

**FIG 1** The relative abundances of microbial communities varied among the treatments at the phylum level. CF, chemical fertilizer; CF_M, chemical fertilizer + manure; CK, no fertilizer; M, manure; SG, sandy grassland. \*\*\**P* < 0.001; \*\**P* < 0.01; and \**P* < 0.05.

decreased (*P* < 0.05) with M and CF_M. The relative abundance of Mrakia increased (*P* < 0.01) with CF, and the relative abundances of Phymatotrichopsis, Phialemonium, and norank_p__Schizoplasmodiida increased (*P* < 0.05) with M and with CF_M (Fig. S5).

Based on the PCoA, the bacterial community at the OTU level displayed a strong separation (*R* = 0.705) along the first two principal coordinates, which together explained 46% of the variance. The fungal community at the OTU level exhibited a moderate separation (*R* = 0.416), with 42% of the variance explained. These results suggest distinct microbial community structures among treatments, with the bacterial community representing a more pronounced separation (Fig. S6). Specific soil microbes are presented in cladograms, with LDA scores exceeding 3, as validated by LEfSe analysis (Fig. 2). Actinobacteriota, Abditibacteriota, Gemmatimonadota, Bacteroidota, Myxococcota, and Bdellovibrionota were enriched in SG; GAL15, Cyanobacteria, NB1-j, MBNT15, unclassified_k_norank_d_Bacteria, Entotheonellaeota, Desulfobacterota, Methylomirabilota, and Latescibacterota were enriched in CK; WPS-2, Armatimonadota, RCP2-54, and Nitrospirota were enriched in CF; and Firmicutes and Fibrobacterota were enriched in M. For soil fungal phyla, SAR_k_norank was enriched in SG; Chytridiomycota, unclassified_d_Eukaryota, and Aphelidea were enriched in CK; Basidiomycota was enriched in CF; and Protosporangiida and Schizoplasmodiida were enriched in M (Fig. 2).

## Soil microbial processes

Enzyme activity related to C acquisition (BG) was greater (*P* < 0.05) in CF_M and CK than in SG (Fig. 3A). The activities of NAG and AP enzymes were greater (*P* < 0.05) in CF_M than in SG (Fig. 3B and D), while that of LAP was greater (*P* < 0.05) in CF_M than in SG and M (Fig. 3C).

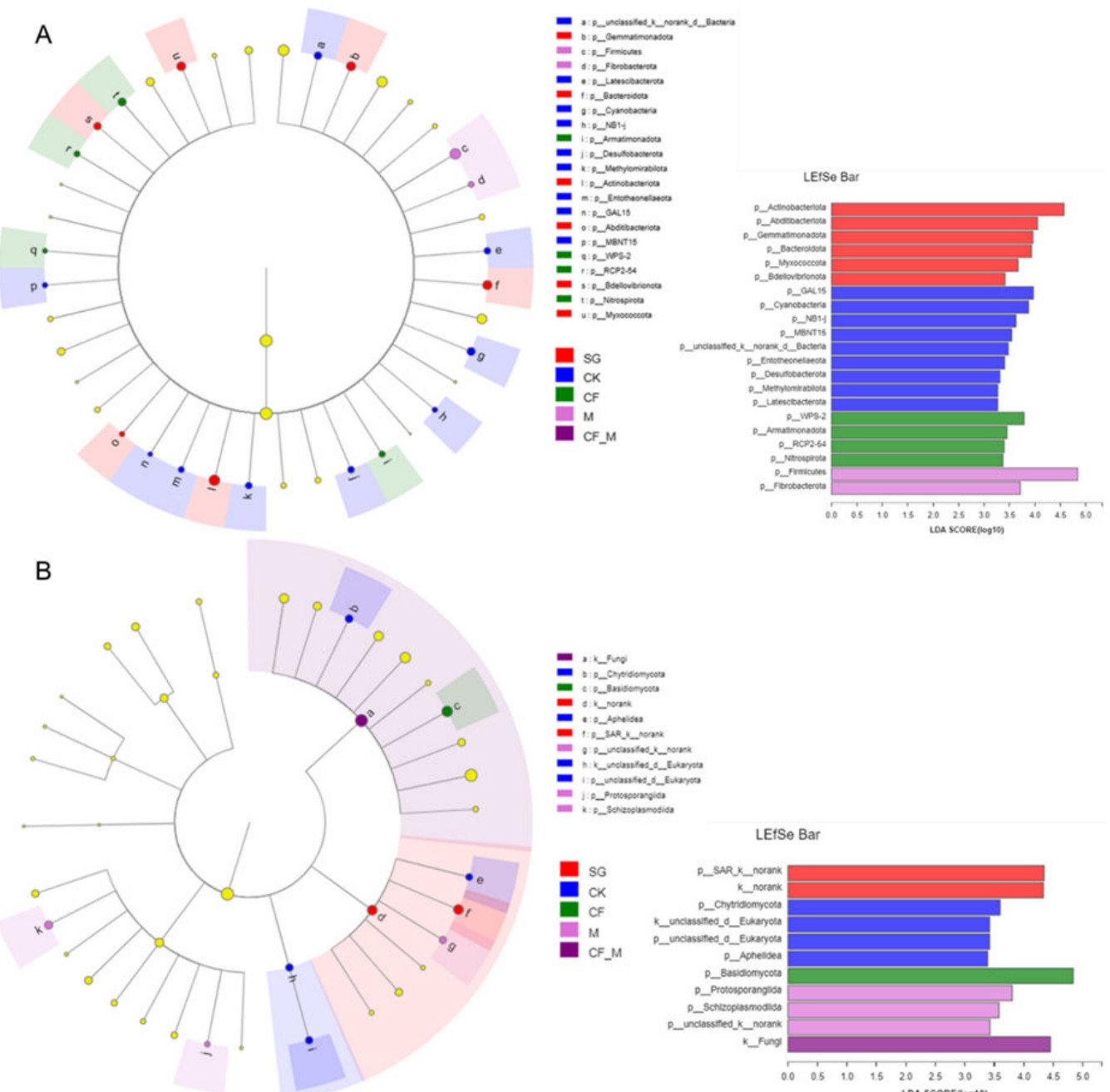

**FIG 2** Cladograms presenting the phylogenetic distribution of the soil microbes from the treatments. Bacterial (A) and fungal (B) biomarkers with LDA scores exceeding 3 in different treatments. CF, chemical fertilizer; CF_M, chemical fertilizer + manure; CK, no fertilizer; M, manure; SG, sandy grassland. Different-colored regions indicate specific soil microbes in treatments (red: SG; blue: CK; green: CF; lilac: M; and purple: CF_M). Concentric circles representing taxonomic hierarchies range from domain to phylum.

The number of functional groups related to nitrogen fixation was greater in CF than in the other treatments (Fig. 4A), while the number of functional groups related to nitrification and denitrification was greater in SG than in the other treatments, except for CF (Fig. 4B and C). The number of functional groups related to degrading cellulose, chitin, and lignin remained constant among treatments (Fig. 4D, E and G), and degrading xylan was greater in M and CF_M than in the other treatments (Fig. 4F). The number of functional groups related to arbuscular mycorrhizal was greater in CK than in the other

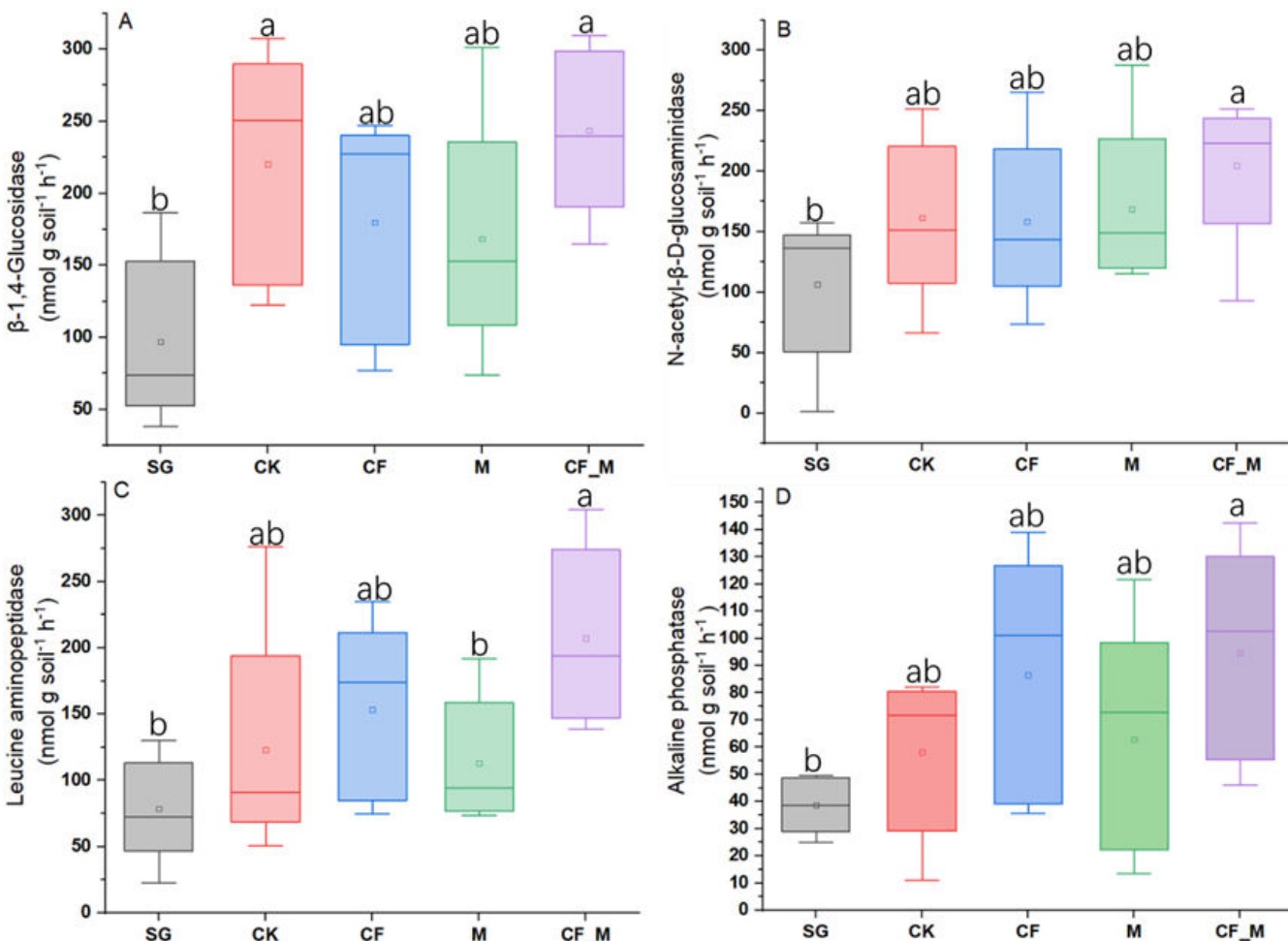

**FIG 3** Soil enzyme activities of β-1,4-glucosidase (A), N-acetyl-β-D-glucosaminidase (B), leucine aminopeptidase (C), and alkaline phosphatase (D) in sandy grassland with different fertilizer treatments. CF, chemical fertilizer; CF_M, chemical fertilizer + manure; CK, no fertilizer; M, manure; SG, sandy grassland. Means with different letters in each soil extracellular enzyme differ from each other ($P < 0.05$).

treatments, except for SG (Fig. 4H), while the number of functional groups related to animal pathogens was greater in CF_M than in the other treatments (Fig. 4I).

## Response of soil microbial processes to environmental factors

Soil C-acquiring enzyme (BG) was correlated positively with SWC, AGB, 18SQ, and the bacterial and fungal Chao indices; N-acquiring enzyme (NAG) was correlated positively with AGB and 18SQ; LAP was correlated positively with SC, TN, AVP, AVN, AGB, 18SQ, and the bacterial Chao index; and P-acquiring enzyme (AP) was correlated positively with SWC, SC, AGB, 18SQ, and the fungal Chao index. Functional groups related to nitrogen fixation were correlated negatively with SC, TN, AVN, and EC; groups related to nitrification were correlated negatively with 18SQ and the bacterial and fungal Chao indices, and groups related to denitrification were correlated negatively with pH and SWC. Functional groups related to degrading chitin were correlated positively with AGB, 16SQ, and 18SQ; groups related to degrading xylan were correlated positively with SWC, EC, SC, TN, AVP, AVN, AGB, 16SQ, 18SQ, and the fungal Chao index, but negatively with SBD; and groups related to degrading lignin enzymes were correlated negatively with pH. Functional groups related to arbuscular mycorrhiza were correlated negatively with SWC, EC, SC, TN, AVP, AVN, AGB, NAG, AP, 16SQ, and 18SQ, and groups related to animal pathogens were correlated positively with SWC, SC, TN, AVN, 18SQ, and fungal Chao index, but negatively with SBD (Fig. 5).

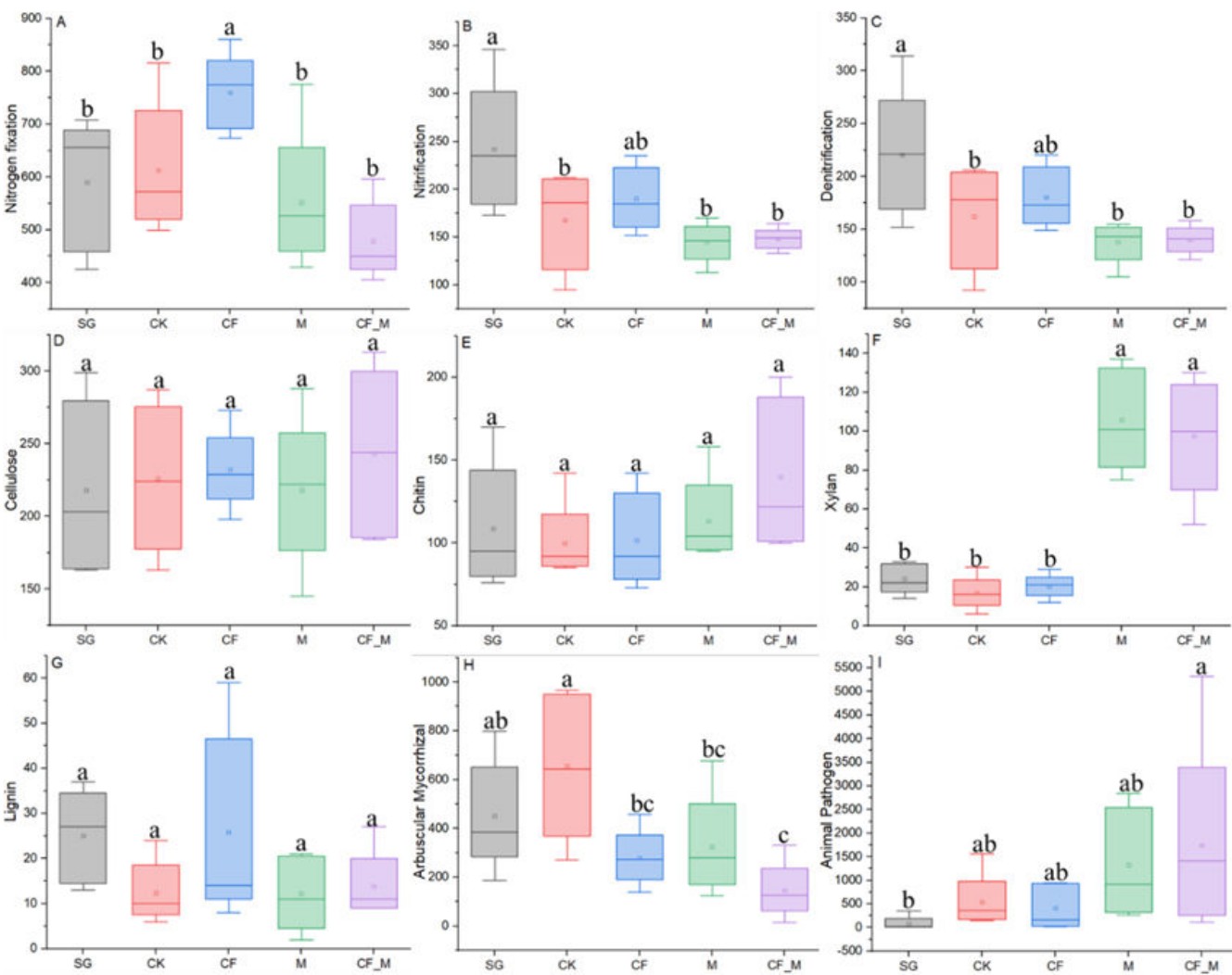

**FIG 4** Predicted functional category groups related to the C (D–H) and N (A–C) cycles and pathogens (I). CF, chemical fertilizer; CF_M, chemical fertilizer + manure; CK, no fertilizer; M, manure; SG, sandy grassland. Means with different letters in each predicted functional group differ from each other ($P < 0.05$).

SEM was used to determine the effect of fertilization, soil properties, and the microbial community on the predicted functional category groups related to C and N cycles (Fig. 6). Soil bulk density (SBD), total soil nitrogen (TN) and carbon (SC), the relative abundance of Firmicutes, and bacterial Chao index (Bchao) explained 57% of the variance of the C cycle functional groups (Fig. 6A). Fertilizer exerted a negative indirect effect on C cycle functional groups (−0.33, by SBD) and a positive direct effect on Bchao and the relative abundance of Firmicutes. Moreover, fertilizer had a positive indirect effect on TN (0.44 through Firmicutes; 0.22 through SBD), SC (0.07 through Bchao; 0.41 through Firmicutes and TN) and C cycle functional groups (0.12 through SBD and TN; 0.11 through SBD, TN and SC; 0.42 through Firmicutes; 0.24 through Firmicutes and TN; 0.22 through Firmicutes, TN and SC; 0.04 through Bchao and SC). TN had positive direct (0.55) and indirect (0.47, through SC) effects on C cycle functional groups, while Firmicutes had positive direct (0.54) and indirect (0.31 by TN; 0.29 by TN and SC) effects on C cycle functional groups. These findings suggest that fertilizer-induced reductions in SBD, coupled with increases in TN, SC, the relative abundance of Firmicutes, and Bchao, collectively enhanced C cycle functional groups.

Based on the final SEM, SBD, TN, SC, the relative abundance of Nitrospirota and Bchao explained 34% of the variance in N cycle functional groups (Fig. 6B). Fertilizer exerted a

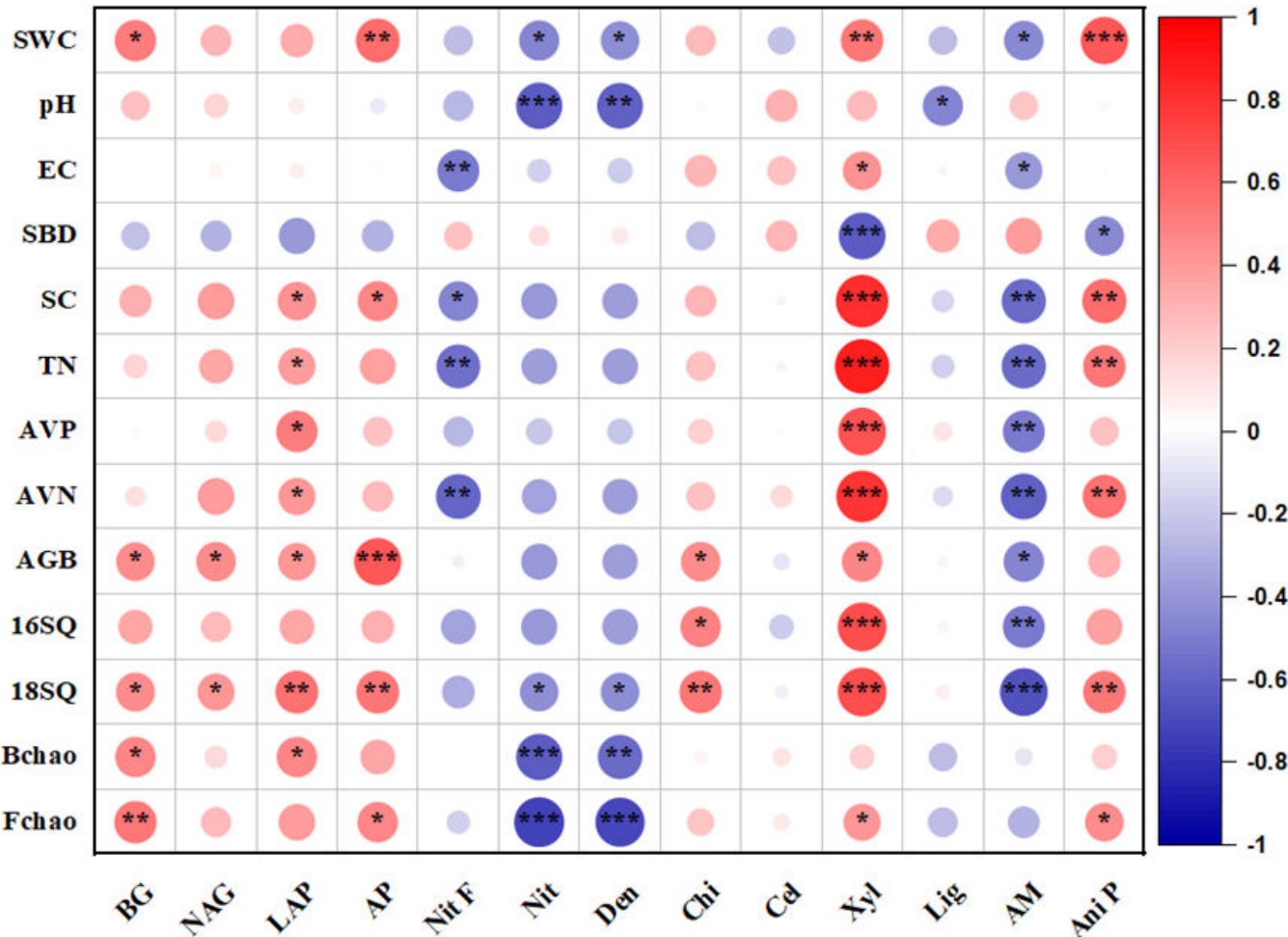

**FIG 5** Correlation coefficients between enzymes, potential functional category groups and soil physico-chemical characteristics, aboveground biomass, microbial absolute abundances, and microbial richness (***$P < 0.001$; **$P < 0.01$; and *$P < 0.05$). Soil water content (%) (SWC); 16SQ, 16S quantity; 18SQ, 18S quantity; AGB, aboveground biomass; AM, arbuscular mycorrhizal; Ani PP, animal pathogen; AVN, available nitrogen; AVP, available phosphorus; Bchao, bacterial Chao index; Cel, cellulose; Chi, chitin; Den, denitrification; EC, electrical conductivity; Fchao, fungal Chao index; Lig, lignin; Nit., nitrification; Nit F, nitrogen fixation; SBD, soil bulk density; SC, total soil carbon; TN, total nitrogen; Xyl, xylan.

negative indirect effect on N cycle functional groups (−0.16 through SBD and TN; −0.20 through TN and SC; and −0.04 through Bchao and SC), a positive direct effect on TN and Bchao, and a positive indirect effect on TN (0.27 through SBD) and SC (0.4 through TN; 0.08 through Bchao). TN had a negative direct effect (−0.58) and a negative indirect effect (−0.47 through SC) on N cycle functional groups, while Nitrospirota had a negative indirect effect (−0.03 through Bchao and SC) on N cycle functional groups. These findings indicate that fertilizer-induced reductions in SBD and the relative abundance of Nitrospirota, and increases in TN, SC, and Bchao, collectively led to a decline in N cycle functional groups.

## DISCUSSION

### Variations of soil microbial structures and processes in response to fertilizers

The absolute abundance of microbial communities could represent their biomasses (40). In the present study, the application of manure or in combination with chemical fertilizer led to an increase in soil bacterial and fungal biomasses. Therefore, organic fertilizer or organic plus inorganic fertilizer improved soil C and N, which were beneficial for microbial growth (41, 42). Moreover, crop residues supply nutrients to soil microbes (43).

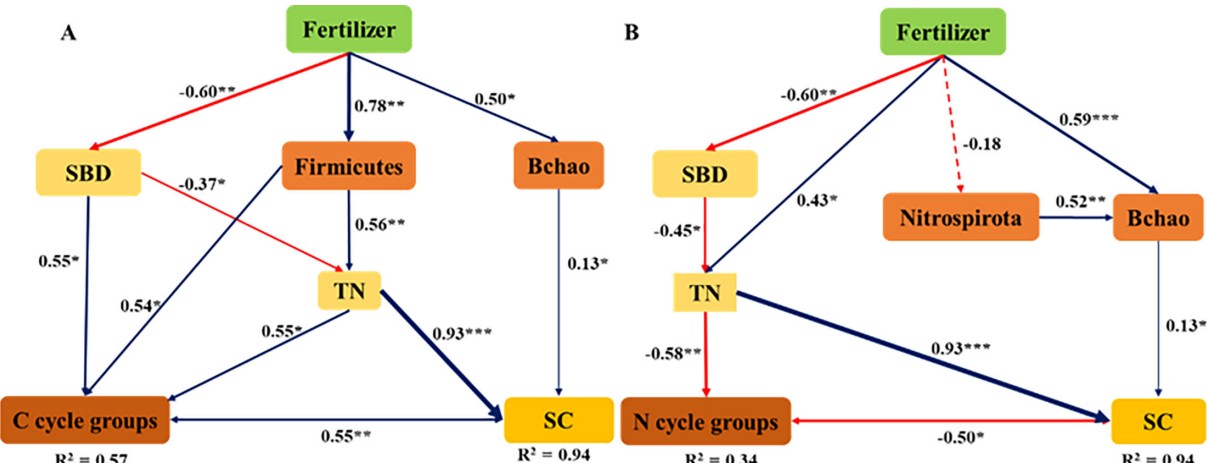

**FIG 6** Structural equation models (SEMs) with soil properties, microbial communities, and predicted functional category groups associated with C cycle (A) and N cycle (B). The thickness of the line represents the strength of the causal relationship, and the numerical values are the path coefficients (***$P < 0.001$; **$P < 0.01$; and *$P < 0.05$). The $R^2$ adjacent to the rectangles indicates the percentage of variance accounted for by the variables in the model. Blue, red, and dotted arrows denote positive, negative, and non-significant influences. The $\chi^2$ value and associated $P$ value were used to assess the model's fit ([A] $\chi^2 = 14.16$, $P = 0.29$, df = 12; [B] $\chi^2 = 16.21$, $P = 0.30$, df = 14). Bchao, bacterial Chao index; C cycle groups, predicted functional category groups associated with C cycling; Firmicutes, relative abundance of Firmicutes; N cycle groups, predicted functional category groups associated with N cycling; Nitrospirota, relative abundance of Nitrospirota; SBD, soil bulk density; SC, total soil carbon; TN, total nitrogen.

The richness and diversity of soil microbial communities play critical roles in sustaining the functionality and health of soil (44, 45). Soil microbial Chao richness increased in response to reclamation, which aligned with findings from an earlier study (46). In contrast, fertilization did not alter soil microbial Chao richness or the bacterial Shannon diversity index. The fungal Shannon diversity index decreased with chemical fertilizer plus manure in this study, contrasting with findings from a long-term study in Kenya where farmyard manure, with or without chemical fertilizer, enhanced both fungal and bacterial diversities (47). These results suggest that fungal diversity exhibits greater sensitivity to the combined organic and inorganic fertilizer than soil bacterial diversity after conversion of sandy grassland to maize production in the arid and semiarid regions.

Fertilization had a notable impact on the soil microbial community structure of the reclaimed sandy grassland producing maize, which supported our hypothesis. This occurred due to the shift in plant composition. Conversion from sandy grassland to maize cropland altered the rhizosphere soil, which is a pivotal driver in shaping the microbial community (48). The shift of vegetation may be responsible for enrichment of new crop rhizosphere microbial communities and the reduction of some native microbial groups in the sandy grassland, as was supported by the LEfSe analysis. The increase of nutrient availability may switch the relationships from symbiotic to competitive among soil microbial species (49). An increase in the ratio of copiotrophic to oligotrophic microbiota may be another explanation for the different responses of bacteria to reclamation and fertilization. Copiotrophs are fast-growing and thrive in environments of abundant nutrients, whereas oligotrophs grow more slowly and are specialized for survival in nutrient-limited conditions (50).

In the current study, Proteobacteria and Actinobacteria emerged as the predominant bacteria phyla, which was consistent with a report in the drylands in Mexico (51). The relative abundance of Firmicutes increased with manure fertilizer and, to a lesser degree, with manure plus chemical fertilizer, which was consistent with earlier findings (52, 53). Firmicutes is known for its rapid response to labile carbon sources and its robust metabolic capabilities (54), and these characteristics could explain its increase with organic fertilizer or organic plus inorganic fertilizer. Bacteroidetes, Gemmatimondetes, and Cyanobacteria are oligotrophic bacteria (55) and, therefore, their relative

abundances were reduced with the application of manure or manure plus chemical fertilizer. Reclamation enhanced the relative abundance of Nitrospirota and reduced the relative abundance of Actinobacteria. The high nutrient availability due to fertilization in the maize cropland led to a greater relative abundance of Nitrospirota than in the control sandy grassland (56). For fungi, the relative abundances of Chytridiomycota and Schizoplasmodiida varied widely among treatments, contrasting with the results of Hu et al. (57), who reported that the relative abundance of Chytridiomycota was not affected by reclamation. The discrepancies between studies may stem from variations in root-microbiota interactions influenced by the site-specific soil environment (58).

In earlier studies, fertilization induced alterations in soil extracellular enzyme activities, and C and N cycling groups were associated closely with productivity and soil C dynamics (14, 59). In turn, changes in productivity and soil C storage affected the supply of substrate and energy for soil microorganisms via residues and root exudates, consequently altering soil microbial communities and soil C and N cycling processes (7, 60).

In the present study, chemical fertilizer plus manure increased the secretions of BG, NAG, LAP, and AP, which indicates that the combination of organic and inorganic fertilizers improved soil microbial growth and modified C and N turnovers in the sandy grassland. The groups related to animal pathogens were also enriched with chemical fertilizer plus manure, which could be detrimental to the soil ecosystem. Nitrospirota generates nitrate ions, which supply substrates for denitrification by complete ammonia oxidation or nitrification. Nitrate contamination has emerged as a worldwide concern, as it pollutes groundwater or remains in the soil (61, 62). Functional groups related to N cycling can exert a positive influence on denitrification and further strengthen the N cycle. Reclamation reduced the abundance of microorganisms associated with nitrification and denitrification, potentially disrupting the N cycle because of the exogenous nutrient addition. However, the results for functional prediction should be viewed with caution. First, the databases for FAPROTAX and FUNGuild are selected from published studies and reports, but coverage across microbial lineages is uneven. Second, the FAPROTAX database is static. Furthermore, the two databases infer functions indirectly from taxonomy and not from direct genetic evidence (metagenomics) or activity measurements. Studies focusing on strain traits or metagenome sequence of these microorganisms in the sandy ecosystem are lacking (45). Systematic and comprehensive studies are needed to manipulate microorganisms for developing a sustainable sandy grassland ecosystem.

## Linkage of soil microbial communities, processes, and effect factors

Soil pH has emerged as a key factor influencing microbial growth and soil C and N cycling, including soil C and N cycling functional groups (45, 63). Microbes prefer neutral conditions, as both acidic and alkaline environments hinder their growth and reproduction (64). In the current study, soil pH decreased with chemical fertilizers, which indicated that the soil acidified (65). However, soil pH elevated with chemical fertilizer plus manure or only manure, with a concomitant increase in soil C and N contents, consistent with earlier findings (46). Most of the increased soil N can be linked to the N input from fertilizers, with a smaller contribution stemming from C accumulation. Soil moisture, electrical conductivity, pH, and C, N, and phosphorus contents were altered by the land use change and management regime, which influenced the diversity and structural composition of microorganisms (66, 51, 57).

Soil C and N contents, SBD, microbial absolute abundance, fungal richness, and aboveground biomass were the key factors influencing the extracellular enzymes in the current study, which is in agreement with earlier reports that soil N and C availability exert a stronger influence on microbial growth than pH in grassland and cropland (67). The functional groups in the present study related to nitrification and denitrification and degrading lignin were correlated negatively with soil pH, aligning with the results

of Dai et al. (63). These results indicated that a lower pH promotes soil C degradation, nitrification, and denitrification.

Plant biomass serves as the main substrate for microbes, thereby increasing the absolute abundance of soil microbiota and bacterial diversity (68, 69). The arbuscular mycorrhiza groups decreased, whereas groups related to degrading chitin and xylan increased with an increase in aboveground biomass, which may be due to the distinct nutrient acquisition capabilities between soil bacteria and fungi (70). Alterations in SBD, soil chemical characteristics, bacterial composition, and Chao richness due to fertilization in the sandy grassland enhanced C cycle and reduced N cycle functional groups, which supported our hypothesis.

The positive correlation between fungal richness and soil respiration suggests that fungi can degrade available and complicated substrates and supports the critical role of fungi in regulating soil functions and C cycling (71–73). Reclamation of sandy grassland to maize production without fertilizer altered the soil microbial Chao richness (SG-CK), whereas 5 years of fertilization did not alter Chao richness in the present study. However, in a 31-year field study in Kenya, organic and inorganic fertilizers altered the soil microbial diversities (47). Earlier reports identified soil microbial biomass, diversity, and taxa as key predictors of microbial respiration, soil C dynamics, and N cycling in marine and terrestrial ecosystems (9, 74, 75), which aligned with the current study that soil microbial biomass and diversity have strong relationships with predicted functional category groups related to soil C and N cycling. Reclamation of sandy grassland and fertilizer increased AGB by more than seven times, indicating that reclamation and fertilization can be a profitable option. There is a lack of direct evidence that reclamation of sandy grassland and fertilization could affect soil C dynamics, nutrient availability, and health status by altering functional category groups related to the C and N cycles and pathogens. Further metagenomic studies on the impact of land use change and fertilization on associated functional communities are warranted for predicting soil C dynamics and nutrient contents in sandy grassland.

## Conclusion

The conversion of sandy grassland to maize cultivation altered soil microbial diversities and community structure, soil extracellular enzyme activities, and predicted functional category groups in the agro-pastoral ecotone of northern China. Reclamation of sandy grassland increased soil microbial Chao richness but did not alter the Shannon diversity index. Reclamation of grassland and fertilization increased AGB by 7–17 times, with chemical fertilizer plus manure providing the best results. The sandy soil microbial functional category groups were driven mainly by SBD, soil chemical characteristics, and bacterial diversity and community structure. The groups related to C cycling were enhanced by fertilization, while the groups related to N cycling were reduced by fertilization. Overall, reclamation and fertilizer improved the soil environment and altered the microbial community structure and functional processes. The results indicated that variations in the microbial community composition driven by reclamation and fertilization in sandy grassland ecosystems could lead to the reduction of soil C and insufficient soil N by altering functional category groups, which could potentially contribute to grassland degradation. These findings are essential for predicting soil carbon sequestration and nutrient cycling in reclaimed sandy grassland.

### ACKNOWLEDGMENTS

The research was supported by the National Natural Science Foundation of China (42207538), the Science and Technology Project of Gansu Province (21JR7RA065), the "Open bidding for selecting the best candidates" Major Demonstration Engineering Project for Scientific and Technological Innovation in Inner Mongolia Autonomous Region (2024JBGS0020), and the Science and Technology Poverty Alleviation Project of Chinese Academy of Sciences (KFJ-FP-202104).

## AUTHOR AFFILIATIONS

[1]Inner Mongolia Naiman Agroecosystem National Field Observation and Research Station, State Key Laboratory of Ecological Safety and Sustainable Development in Arid Lands, Northwest Institute of Eco-Environment and Resources, Chinese Academy of Sciences, Lanzhou, China

[2]Urat Desert-grassland Research Station, Northwest Institute of Eco-Environment and Resources, Chinese Academy of Sciences, Lanzhou, China

[3]University of Chinese Academy of Sciences, Beijing, China

[4]Key Laboratory of Stress Physiology and Ecology in Cold and Arid Region of Gansu Province, Lanzhou, China

[5]Blaustein Institutes for Desert Research, Ben-Gurion University of Negev, Be'er Sheva, Israel

[6]College of Ecology, Lanzhou University, Lanzhou, China

## AUTHOR ORCIDs

Rui Zhang   http://orcid.org/0000-0003-1520-3436
Yuqiang Li   http://orcid.org/0000-0001-5264-8122

## AUTHOR CONTRIBUTIONS

Rui Zhang, Data curation, Formal analysis, Funding acquisition, Investigation, Software, Visualization, Writing – original draft, Writing – review and editing | Yulin Li, Conceptualization, Methodology, Project administration, Validation | Xueyong Zhao, Methodology, Resources, Writing – review and editing | A. Allan Degen, Validation, Writing – review and editing | Xinping Liu, Data curation, Investigation, Resources | Jie Lian, Investigation, Methodology, Software, Visualization | Yuqiang Li, Resources, Supervision, Validation, Writing – review and editing | Yalin Wu, Methodology, Software, Writing – review and editing | Zhanhuan Shang, Validation, Writing – review and editing

## DATA AVAILABILITY

The raw sequencing data generated from this study are deposited in the Genome Sequence Archive in the BIG Data Center (https://bigd.big.ac.cn/) (accession number CRA037529). All other relevant data from this study are available from the senior author upon reasonable request.

## ADDITIONAL FILES

The following material is available online.

### Supplemental Material

**Supplemental tables and figures (Spectrum02963-25-S0001.doc).** Tables S1 and S2, and Figures S1 to S6.

### Open Peer Review

**PEER REVIEW HISTORY (review-history.pdf).** An accounting of the reviewer comments and feedback.

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
