## [Reviewer comments · Microbiology Spectrum]

Microbiology Spectrum

Five-year fertilization alters soil microbial composition and functionality in sandy grassland

Rui Zhang, Yulin Li, Xueyong Zhao, Allan Degen, Xinping Liu, Jie Lian, and Yuqiang Li

Corresponding Author(s): Rui Zhang, Chinese Academy of Sciences Northwest Institute of Eco-Environment and Resources

Review Timeline:

Submission Date:	September 17, 2025
Editorial Decision:	November 23, 2025
Revision Received:	December 16, 2025
Editorial Decision:	January 9, 2026
Revision Received:	February 8, 2026
Accepted:	February 27, 2026

Editor: Katharina Kujala

Reviewer(s): Disclosure of reviewer identity is with reference to reviewer comments included in decision letter(s). The following individuals involved in review of your submission have agreed to reveal their identity: Saidu Abdullahi (Reviewer #2)

Transaction Report:

DOI: <https://doi.org/10.1128/spectrum.02963-25>

The manuscript by Zhang et al. describes an investigation of soil microbial community structure and soil biogeochemistry following conversion of a sandy grassland to farmland. They investigated 5 treatments, including background sandy grassland and combinations of converted land with different fertilizers. They use 16S and presumably ITS amplicon sequencing to study the composition and structures of the microbial communities, and qPCR to estimate 16S and 18S rRNA gene copy numbers. They paired this data with soil enzyme assays and soil geochemistry measures. There were differences in microbial community structures by fertilization regime. Enzymes and microbial biomass increased with fertilization. Putative nitrogen cycling taxa decreased with fertilization while putative carbon cycling taxa increased. Overall, this manuscript presents a complex set of data illustrating how fertilizer impacts microbial community structure. It will be of interest to other researchers focusing on sandy soils, however it is sparse on actual descriptions of the microbial communities.

Major Concerns:

1. Lack of data availability. The authors must include a data availability statement, deposit their 16S and ITS or 18S reads, and provide all processed data files as supplemental files for reproducibility and transparency. This includes the OTU table with taxonomy and the FAPROTAX and FUNGuild outputs.
2. Missing critical methodological information about amplicon sequencing processing. First, what primers were used? A reference to another paper is not appropriate because it is the central data of this paper. Second, no details about processing in mothur are provided. Is this paper using ASVs or OTUs? If OTUs, what identity level were sequences grouped at to assign OTUs? How many reads are there per sample, and were they subsampled to a common depth? The authors should increase their methodological reporting. If they used ASVs, they should not be using the Chao1 index- see this paper: <https://doi.org/10.1093/ismejo/wrae106>.
3. There is very little discussion on the microbial composition of these soils. The authors are not really showing a “shift in microbial composition” as their title states. Figure S3 shows phylum-level bar charts of the composition, but they show no changes in overall composition. Figure 1 presumably focuses on the phyla that are different across treatments, but it appears the changes in relative abundance are largely driven by the increase in Firmicutes in M and CF_M. While they have some discriminant taxa identified by Lefse in figure 2, it is unclear how abundant these taxa are in the community. The authors should consider including beta-diversity analyses and analyses of taxa at finer levels of taxonomy to more thoroughly investigate shifts in composition.

Minor concerns

1. line 322-323: I would like the authors to expand on the limitations of FAPROTAX more than saying they are “not reliable sometimes”
2. This manuscript is readable, but could benefit from refinement of the English language.
3. Figures 3-4, Fig S1: I would like to see the individual datapoints overlaid on the boxplots for transparency.
4. Figure 6: please define the abbreviations in the image.
5. Figure S3: is this the average? Could you provide a barplot for every replicate to show variation across replicates?

To Katharina Kujala

Editor

Microbiology Spectrum

We are very grateful for your efforts and for the reviewers' constructive suggestions in reviewing our manuscript, "Five-year fertilization alters soil microbial composition and functionality in sandy grassland" (ID: **Spectrum02963-25**). Please find our point-by-point responses to each comment below. We hope that you will find that our revised manuscript addressed the issues raised by the reviewers satisfactorily and that the paper is now in proper form for publication in *Microbiology Spectrum*.

Reviewer #1 (Comments for the Author):

The manuscript by Zhang et al. describes an investigation of soil microbial community structure and soil biogeochemistry following conversion of a sandy grassland to farmland. They investigated 5 treatments, including background sandy grassland and combinations of converted land with different fertilizers. They use 16S and presumably ITS amplicon sequencing to study the composition and structures of the microbial communities, and qPCR to estimate 16S and 18S rRNA gene copy numbers. They paired this data with soil enzyme assays and soil geochemistry measures. There were differences in microbial community structures by fertilization regime. Enzymes and microbial biomass increased with fertilization. Putative nitrogen cycling taxa decreased with fertilization while putative carbon cycling taxa increased. Overall, this manuscript presents a complex set of data illustrating how fertilizer impacts microbial community structure. It will be of interest to other researchers focusing on sandy soils, however it is sparse on actual descriptions of the microbial communities.

Response: We revised the manuscript based on the reviewer's constructive suggestions. More descriptions of the microbial communities were added in the revised version. Point-by-point responses to the comments are presented below.

Major Concerns:

1. Lack of data availability. The authors must include a data availability statement, deposit their 16S and ITS or 18S reads, and provide all processed data files as supplemental files for

reproducibility and transparency. This includes the OTU table with taxonomy and the FAPROTAX and FUNGuild outputs.

Response: All data from this study are available from the senior author upon reasonable request. The raw reads have been uploaded to the Genome Sequence Archive in the BIG Data Center (<https://bigd.big.ac.cn/>). The data files of OTU table with taxonomy, the FAPROTAX and FUNGuild outputs are provided as supplemental files.

2. Missing critical methodological information about amplicon sequencing processing. First, what primers were used? A reference to another paper is not appropriate because it is the central data of this paper. Second, no details about processing in mothur are provided. Is this paper using ASVs or OTUs? If OTUs, what identity level were sequences grouped at to assign OTUs? How many reads are there per sample, and were they subsampled to a common depth? The authors should increase their methodological reporting. If they used ASVs, they should not be using the Chao index- see this paper: <https://doi.org/10.1093/ismejo/wrae106>.

Response: The methodological information was added in the section **2.4 Sequencing, and bioinformatics analysis in the revised version.**

2.4 Sequencing, and bioinformatics analysis

The Mag-Bind® Soil DNA Kit (Omega Bio-tek, Norcross, GA, USA) was used to extract soil genomic DNA. The concentration and the purity of the extracted DNA were determined by the 260/280 nm ratio (1.8 to 2.2) using a NanoDrop 2000 UV-vis Spectrophotometer (Thermo Scientific, Wilmington, DE, USA). Absolute abundances (the gene copy numbers) of the bacterial and fungal communities were estimated using quantitative real-time polymerase chain reaction (qPCR) amplification techniques with a Line-Gen 9600 Plus Cycler (Thermo Fisher Scientific Inc., Waltham, MA, USA). PCR amplification used the universal primer 338F-806R for bacteria (Mohd Yusoff et al., 2013) and SSU0817F-1196R for fungi (Adams et al., 2013) with a GeneAmp 9700 PCR system

(Applied Biosystems, Foster City, CA, USA). PCR products were quantified using the QuantiFluor™-ST Fluorometer (Promega Biotech, Beijing, China), and sequencing (by Majorbio Company, Shanghai, China) used an Illumina MiSeq platform (San Diego, USA). The raw reads of sequencing were quality-filtered by Fastp, merged by FLASH and chimeric sequences were removed (Magoc & Salzberg, 2011). To ensure comparability in diversity analyses, all samples were subsampled to an even depth of 40,000 sequences (Figure S1) per sample in Mothur. Sequences were grouped into the same operational taxonomic units (OTUs) with high similarity ($\geq 97\%$) using UPARSE version 11 (Edgar, 2013). The representative sequences for each OTU were annotated and classified based on the SILVA database (<http://www.arb-silva.de>) using the RDP (Ribosomal Database Project) classifier (<http://rdp.cme.msu.edu/>). The functional genes were predicted by FAPROTAX for bacteria and FUNGuild for fungi using the Kyoto Encyclopedia of Genes and Genomes (KEGG) (<https://www.kegg.jp/kegg/>) and National Center for Biotechnology Information Non redundant (NCBI NR) (<https://www.ncbi.nlm.nih.gov/>) databases (Nguyen et al., 2016; Zheng et al., 2019) based on OTU representative sequences.

Figure S1 The rarefaction curves of soil microbiota in the samples. SG = Sandy grassland; CK=

No fertilizer; CF = Chemical fertilizer; M = Manure; CF_M = Chemical fertilizer + Manure.

3. There is very little discussion on the microbial composition of these soils. The authors are not really showing a "shift in microbial composition" as their title states. Figure S3 shows phylum-level bar charts of the composition, but they show no changes in overall composition. Figure 1 presumably focuses on the phyla that are different across treatments, but it appears the changes in relative abundance are largely driven by the increase in Firmicutes in M and CF_M. While they have some discriminant taxa identified by Lefse in figure 2, it is unclear how abundant these taxa are in the community. The authors should consider including beta-diversity analyses and analyses of taxa at finer levels of taxonomy to more thoroughly investigate shifts in composition.

Response: Analyses of taxa at genus levels and the dissimilarity of microbial community at the OTU level with different fertilizer treatments were added in the section 3.2 Soil microbiome composition in the revised version.

The relative abundances of Gemmatimonadaceae and Rubrobacter increased ($p < 0.001$) with reclamation, but of Knufia and Verticillium decreased ($p < 0.05$) with reclamation and fertilizer. The relative abundances of Sporosarcina, Paenibacillus, Steroidobacter and Turcibacter increased ($p < 0.05$) but of Geodermatophilaceae, Rubrobacter and Nitrospira decreased ($p < 0.05$) with M and CF_M. The relative abundance of Mrakia increased ($p < 0.01$) with CF, and the relative abundances of Phymatotrichopsis, Phialemonium and norank_p_Schizoplasmodiida increased ($p < 0.05$) with M and with CF_M (Figure S5).

Based on the PCoA, the bacterial community at the OTU level displayed a strong separation ($R = 0.705$) along the first two principal coordinates, which together explained 46% of the variance. The fungal community at the OTU level exhibited a moderate separation ($R =$

0.416), with 42% of the variance explained. These results suggest distinct microbial community structures among treatments, with the bacterial community representing a more pronounced separation (Figure S6).

Figure S5 Relative abundances of soil bacteria (left) and fungi (right) among treatments at the genus level. SG = Sandy grassland; CK = No fertilizer; CF = Chemical fertilizer; M = Manure; CF_M = Chemical fertilizer + manure. *** $p < 0.001$; ** $p < 0.01$; * $p < 0.05$.

Figure S6 The dissimilarity of bacteria (left) and fungi (right) at the OTU level with different fertilizer treatments. SG = Sandy grassland; CK = No fertilizer; CF = Chemical fertilizer; M = Manure; CF_M = Chemical fertilizer + manure.

Minor concerns

1. line 322-323: I would like the authors to expand on the limitations of FAPROTAX more than saying they are "not reliable sometimes"

Response: The limitations of FAPROTAX were expanded in the revised version.

However, the results for functional prediction should be viewed with caution. Firstly, the databases for FAPROTAX and FUNGuild are selected from published studies and reports but coverage across microbial lineages are uneven and, secondly, the FAPROTAX database is static. Furthermore, the two databases infer functions indirectly from taxonomy and not from direct genetic evidence (metagenomics) or activity measurements.

2. This manuscript is readable, but could benefit from refinement of the English language.

Response: The manuscript was edited carefully by a native English speaker.

3. Figures 3-4, Fig S1: I would like to see the individual datapoints overlaid on the boxplots for transparency.

Response: The individual datapoints overlaid on the boxplots were included in the revised Figures 3-4, Fig S2.

Figure 3

Figure 4

Fig S2

4. Figure 6: please define the abbreviations in the image.

Response: The abbreviations in the figure 6 were defined in the revised version.

Figure 6 Structural equation models (SEMs) with soil properties, microorganism communities, and predicted functional category groups associated with C cycle (A) and N cycle (B). The thickness of the line represents the strength of the causal relationship, and the numerical values are the path coefficients (** $p < 0.01$; *** $p < 0.001$ and * $p < 0.05$). The R^2 adjacent to the rectangles indicate the percentage of variance accounted for by the variables in the model. Blue, red and dotted arrows denote positive, negative and non-significant influences. The χ^2 value and associated P-value were used to assess the model's fit (A. $\chi^2 = 14.16$, $p = 0.29$, $df = 12$; B. $\chi^2 = 16.21$, $p = 0.30$, $df = 14$). SBD = Soil bulk density; SC = Total soil carbon; TN = Total nitrogen; Bchao = Bacterial Chao index; Firmicutes = the relative abundance of

Firmicutes; Nitrospirota = the relative abundance of Nitrospirota; C cycle groups = predicted functional category groups associated with C cycling, N cycle groups = predicted functional category groups associated with N cycling.

5. Figure S3: is this the average? Could you provide a barplot for every replicate to show variation across replicates?

Response: The data in Figure S4 is the average value of each treatment from 5 replicates.

The microbial community barplot for every replicate

Reviewer #2 (Comments for the Author):

The study was carried out to majorly evaluate the influence of reclamation and fertilization on soil properties and soil microbial diversities and community structure. This is so important to provide insight towards sustainable agricultural activities in sandy soils.

The content is so rich. However, the language structure is somehow poor with lots of grammatical errors. This needs to be addressed to improve the quality of the paper. For instance, the first and second sentences of the abstract need to be polished before real meaning is obtained.

Response: The manuscript was edited carefully by a native English speaker. The first and second sentences of the abstract were re-written in the revised version.

The impacts of reclamation and fertilization on soil microbial communities and functional groups related to carbon (C) and nitrogen (N) cycling in sandy grassland are not well understood. To fill this gap, three types of fertilizers, namely, chemical fertilizer (CF), manure (M) and chemical fertilizer plus manure (CF_M), were applied annually for five years to reclaimed sandy cropland planted to maize.

Re: Spectrum02963-25R1 (**Five-year fertilization alters soil microbial composition and functionality in sandy grassland**)

Dear Dr. Rui Zhang:

Thank you for submitting a revised version of your manuscript.

While you have sufficiently addressed most of the reviewers' concerns, some issues still remain. Please take a look at the comments provided by the reviewer and address these accordingly.

Revision Guidelines

Sincerely,
Katharina Kujala
Editor
Microbiology Spectrum

Reviewer #1 (Comments for the Author):

The authors have addressed most of my concerns. However, there is still not a data availability statement. The authors should review the ASM Open Data policy: <https://journals.asm.org/open-data-policy>

The author's reply to me indicates they deposited the reads, but they do not provide any accession numbers in the manuscript.

This must be provided in a data availability statement and/or supplemental table.

I appreciate the inclusion of the OTU tables and FAPROTAX and FUNGuild outputs in the supplemental. However, the OTU tables appear to be the raw OTU tables, not the subsampled to 40000 ones described in the text. Those should be provided too. Additionally, the sample names in these tables are not unique, and there is no way to link them to public data via an accession.

To Katharina Kujala

Editor

Microbiology Spectrum

We are very grateful for your efforts and for the reviewers' constructive suggestions in reviewing our manuscript, "Five-year fertilization alters soil microbial composition and functionality in sandy grassland" (ID: **Spectrum02963-25R1**). Please find our point-by-point responses to each comment below. We hope that you will find that our revised manuscript addressed the issues raised by the reviewers satisfactorily and that the paper is now in proper form for publication in *Microbiology Spectrum*.

Reviewer #1 (Comments for the Author):

The authors have addressed most of my concerns. However, there is still not a data availability statement. The authors should review the ASM Open Data policy: <https://journals.asm.org/open-data-policy>

Response: The data availability statement is provided in the revised version.

Data availability

The raw sequencing data generated from this study are deposited in the Genome Sequence Archive in the BIG Data Center (<https://bigd.big.ac.cn/>), with accession number **CRA037529**. All other relevant data from this study are available from the senior author upon reasonable request.

The author's reply to me indicates they deposited the reads, but they do not provide any accession numbers in the manuscript. This must be provided in a data availability statement and/or supplemental table.

Response: The accession number CRA037529 is provided in a data availability statement in the revised version.

The raw sequencing data generated from this study are deposited in the Genome Sequence Archive in the BIG Data Center (<https://bigd.big.ac.cn/>) with accession number

CRA037529.

I appreciate the inclusion of the OTU tables and FAPROTAX and FUNGuild outputs in the supplemental. However, the OTU tables appear to be the raw OTU tables, not the subsampled to 40000 ones described in the text. Those should be provided too. Additionally, the sample names in these tables are not unique, and there is no way to link them to public data via an accession.

Response: **The unique sample names in these tables are provided in the revised version. The OTU tables in this paper are the raw OTU tables. The appropriate description regarding the number of reads were revised in this version.**

2. How many reads are there per sample, and were they subsampled to a common depth?

Response: **The reads per sample are provided in the data files. The methodological information was rewritten in the revised version in section 2.4 Sequencing, and bioinformatics analysis.**

2.4 Sequencing, and bioinformatics analysis

The Mag-Bind® Soil DNA Kit (Omega Bio-tek, Norcross, GA, USA) was used to extract soil genomic DNA. The concentration and the purity of the extracted DNA were determined by the 260/280 nm ratio (1.8 to 2.2) using a NanoDrop 2000 UV-vis Spectrophotometer (Thermo Scientific, Wilmington, DE, USA). Absolute abundances (the gene copy numbers) of the bacterial and fungal communities were estimated using quantitative real-time polymerase chain reaction (qPCR) amplification techniques with a Line-Gen 9600 Plus Cycler (Thermo Fisher Scientific Inc., Waltham, MA, USA). PCR amplification used the universal primer 338F-806R for bacteria (Mohd Yusoff et al., 2013) and SSU0817F-1196R for fungi (Adams et al., 2013) with a GeneAmp 9700 PCR system (Applied Biosystems, Foster City, CA, USA). PCR products were quantified using the

QuantiFluor™-ST Fluorometer (Promega Biotech, Beijing, China), and sequencing (by Majorbio Company, Shanghai, China) used an Illumina MiSeq platform (San Diego, USA). The raw reads of sequencing were quality-filtered by Fastp, merged by FLASH and chimeric sequences were removed (Magoc & Salzberg, 2011). The number of reads for all bacterial samples ranged between 34356 and 45469 and for all fungi samples ranged between 38549 to 49231 (Figure S1). Sequences were grouped into the same operational taxonomic units (OTUs) with high similarity ($\geq 97\%$) using UPARSE version 11 (Edgar, 2013). The representative sequences for each OTU were annotated and classified based on the SILVA database (<http://www.arb-silva.de>) using the RDP (Ribosomal Database Project) classifier (<http://rdp.cme.msu.edu/>). The functional genes were predicted by FAPROTAX for bacteria and FUNGuild for fungi using the Kyoto Encyclopedia of Genes and Genomes (KEGG) (<https://www.kegg.jp/kegg/>) and National Center for Biotechnology Information Non-redundant (NCBI NR) (<https://www.ncbi.nlm.nih.gov/>) databases (Nguyen et al., 2016; Zheng et al., 2019) based on OTU representative sequences.

Re: Spectrum02963-25R2 (**Five-year fertilization alters soil microbial composition and functionality in sandy grassland**)

Dear Dr. Rui Zhang:

Your manuscript has been accepted, and I am forwarding it to the ASM production staff for publication. Your paper will first be checked to make sure all elements meet the technical requirements. ASM staff will contact you if anything needs to be revised before copyediting and production can begin. Otherwise, you will be notified when your proofs are ready to be viewed.

Sincerely,
Katharina Kujala
Editor
Microbiology Spectrum